# Identifying Truthful Inheritance in Family Models and Enhancing Truthfulness

## Abstract

Recent advances in large language models (LLMs) have led to emergence of specialized multimodal LLMs (MLLMs), creating distinct model families that share a common foundation language models. This work investigates whether a core traits like **truthfulness** are inherited along this evolutionary trajectory. To quantify this trait, we employ linear probing on the models' internal representations. Our analysis of Vicuna and Qwen model families reveals a key finding: a strong correlation in truthfulness scores between LLMs and their finetuned MLLMs counterparts, even when they are finetuned or probed with different modalities and datasets. Building on this findings, we propose a soft gating method using the *Truthfulness* score to amplify the influence of these context-truthful heads to improve the context grounding ability while preserving the contributions of other heads. We validate our approach on base LLMs on HaluEval benchmark, demonstrating an improved ability for context truthful reasoning. We then show that the Truthfulness scores obtained from base LLMs can be effectively transferred and applied as a soft gate to its finetuned variants, demonstrating its improved performance on POPE and CHAIR benchmark. The performance gain from this transfer is comparable to that obtained by probing the MLLMs directly, highlighting the potential for a unified approach to enhance truthfulness across an entire model family. Our work demonstrates a novel method for leveraging a model's inherent, inherited traits to systematically improve its truthfulness.

## 1 Introduction

Recent advancements in large language models (LLMs) has given rise to a wide range of specialized models, all of which are originated from a core foundational LLMs. This pattern reflects a broader trend: rather than building entirely new models from scratch, base LLMs are often refined through fine-tuning or multimodal extensions to serve domain-specific needs—ranging from mathematical reasoning to vision-language understanding, or even multi-sensory processing. Such evolutionary trajectories highlight that many advanced multimodal LLMs (MLLMs) share a clear lineage with their base LLMs.

*Do MLLMs inherit the truthfulness trait from their base LLMs? If so, can this inherited characteristic be leveraged to develop a unified method that enhances truthfulness across both base LLMs and their finetuned MLLMs?*

We hypothesize that attention heads vary in the extent to which they encode context-faithful information, and that this degree of context truthfulness can be quantified using the linear probing methodology introduced by ITI (Li et al., 2023b). To examine whether this property is inherited within model families, we analyze correlations of context-truthfulness scores both within and across model lineages under diverse dataset settings. Specifically, we study Vicuna-7B (Chiang et al., 2023) as a base LLM and its fine-tuned counterparts, LLaVA-1.5 (Liu et al., 2024a) and LLaVA-NeXT (Li et al., 2024) as well as Qwen2.5 family (Qwen et al., 2025), including Qwen2.5-VL-Instruct (Bai et al., 2025) and Qwen2.5-VL-Omni (Xu et al., 2025). Our analysis reveals the key property within model families: **Inheritance.** Under single-dataset probing, MLLMs exhibit high correlation with their base LLMs, regardless of their specialization for different modalities such as vision or audio. Moreover, even when LLMs and MLLMs are probed using data from different sources, models

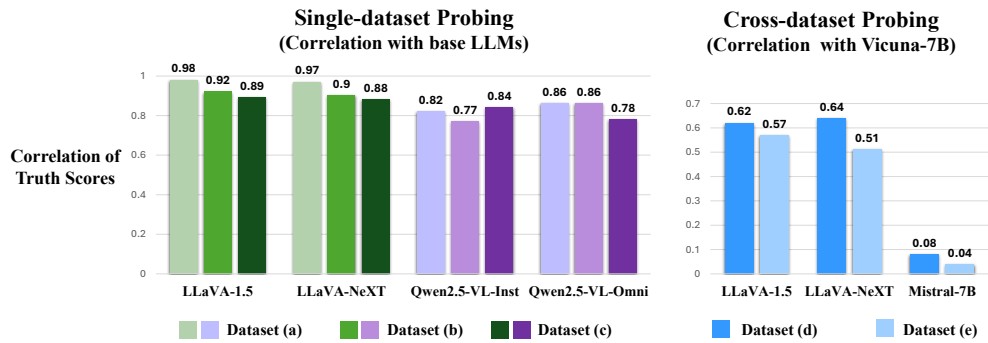

Figure 1: **Correlation of Truth Scores in Single- and Cross-dataset Probing.** *Left: Single-dataset probing results.* Truth Scores for LLaVA-1.5 and LLaVA-NeXT show high correlation with Vicuna-7B (base LLM), and Qwen2.5-VL-Instruct and Qwen2.5-VL-Omni similarly correlate highly with Qwen2.5-7B (base LLM). *Right: Cross-dataset probing results*, where all models are compared against Vicuna-7B. Models from a different family (Mistral-7B) exhibit almost no correlation. For probing dataset setups, please refer to Tab. 1 and Tab. 2.

belonging to the same family maintain substantially higher truthfulness correlations compared to models from unrelated families.

This finding suggests that the truthfulness-related behavior of attention heads is largely preserved when a base LLM is fine-tuned into downstream variants, even when the LLMs and their fine-tuned models are probed using different datasets.

Building on these insights, we propose a Soft Gating strategy that leverages the obtained Truth Scores to amplify the influence of context-truthful heads, thereby ensuring that the model's final outputs are more faithfully grounded in the given context. Importantly, we show that this strategy is not only effective within a single model, but also generalizes consistently across model families sharing the same backbone.

To begin, we validate the obtained truthfulness scores by applying them as a soft gate to the same model and evaluating its ability to assess the faithfulness of the given context. Further, we examine whether the truthfulness scores obtained from base LLMs can function as a soft gate in their fine-tuned variants–including instruction-tuned and multimodal models–and evaluate their performance on the HaluEval, as well as on POPE and CHAIR, which assess hallucination mitigation in MLLMs.

Unlike previous approaches that either rely on model-specific interventions for hallucination reduction or head-level studies that remained descriptive without actionable refinement, our work identifies the inheritance of truthfulness within model lineages and leverages it to improve model truthfulness. By showing that truthfulness scores can be stably inherited and transferred within model families, we establish a principled foundation for refining both LLMs and their finetuned extensions toward greater truthfulness.

Our contributions are summarized as follows:

- **Identifying the Identity of Context-Truthful Heads.** Building on ITI's probing procedure, we measure how well each transformer head grounds responses in the context, yielding a Context-Truthfulness Score (Truth Score).

- **Discovering the Inheritance of Context-Truthful Heads.** Single- and cross-dataset analyses show that Truth Scores are strongly correlated within model families, indicating preservation of context-truthful heads when base LLMs are fine-tuned into LLMs or MLLMs.

- **Soft-Gating for Truthfulness Enhancement.** We propose a soft-gating strategy using Truth Scores to improve model truthfulness, and demonstrate that Truth Scores from base LLMs can be effectively transferred to finetuned LLMs/MLLMs, yielding gains on HaluEval, POPE, and CHAIR.

| | Probing Data | |
|---|---|---|
| | **LLMs** | **MLLMs** |
| (a) | HaluEval | HaluEval text-only |
| (b) | HaluEval | HaluEval w/ black img |
| (c) | PhD-text | PhD-img |

Table 1: **Dataset for Single-dataset Probing.**

| | Probing Data | |
|---|---|---|
| | **LLMs** | **MLLMs** |
| (d) | HaluEval | RLHF-V |
| (e) | PhD-text + HaluEval | PhD-img + RLHF-V |

Table 2: **Dataset for Cross-dataset Probing.**

## 2 IDENTIFYING COMPONENTS FOR CONTEXT-BASED TRUTHFUL REASONING

Recent research has made significant strides in demystifying the internal mechanisms of Large Language Models (LLMs). A particularly compelling line of inquiry suggests that abstract concepts are encoded in interpretable directions within the model's activation space. For example, Li et al. (2023b) introduced Inference-Time Intervention (ITI), a technique that enhances model truthfulness by identifying and shifting activations in specific attention heads. Their findings indicate that models may possess latent "knowledge" of the truth, even when their generated outputs are false.

However, in many real-world applications, truthful reasoning requires more than accessing parametric knowledge—it also depends critically on how well the model leverages the given context. For instance, in Multimodal Large Language Models (MLLMs), tasks such as the widely studied "Where is Wally?" question require accurate grounding in the provided image, rather than relying solely on pre-trained internal knowledge. Motivated by this distinction, we move beyond ITI and focus on identifying attention heads that are not only truthful but also context-referential. Specifically, we aim to characterize and intervene on heads that reliably attend to context in a manner that supports truthful and grounded responses.

### 2.1 PRELIMINARY

Formally, in a Transformer layer $l$, the Multi-Head Attention (MHA) mechanism is composed of $H$ attention heads, each applying an independent linear projection to the residual representation. Given an input $x_l \in \mathbb{R}^d$, the $h$-th head projects it into query, key, and value subspaces via learned matrices $Q_l^h, K_l^h, V_l^h$. The head output is computed as:

$$Att_l^h(x_l) = \text{softmax}\left(\frac{Q_l^h x_l (K_l^h x_l)^\top}{\sqrt{d_k}}\right) V_l^h x_l, \tag{1}$$

where $d_k$ denotes the key dimension. The outputs of all heads are then aggregated through an output projection $W_l^o$ and added back to the residual stream:

$$o_l = W_l^o \cdot \text{Concat}_{h=1}^{H}(Att_l^h(x_l)) \tag{2}$$

$$x_{l+1} = x_l + o_l. \tag{3}$$

This formulation shows that each head contributes a distinct contextual transformation, which is subsequently integrated by the Multi-Layer Perceptron (MLP) through nonlinear operations.

### 2.2 FINDING CONTEXT TRUTHFUL HEAD

As introduced in Sec. 2, evaluating whether a Transformer layer truthfully leverages contextual information is most precise at the granularity of individual attention heads. Each head selectively references tokens from the context and adds its transformed representation into the residual stream. By analyzing heads individually, one can assess whether the contextual information is faithfully preserved or distorted.

We adopt the emerging view that neural networks encode interpretable directions in activation space and hypothesize that certain heads correspond to **truthfulness**. Specifically, we examine whether each head integrates context in a reliable manner or propagates misleading signals. To test this, we

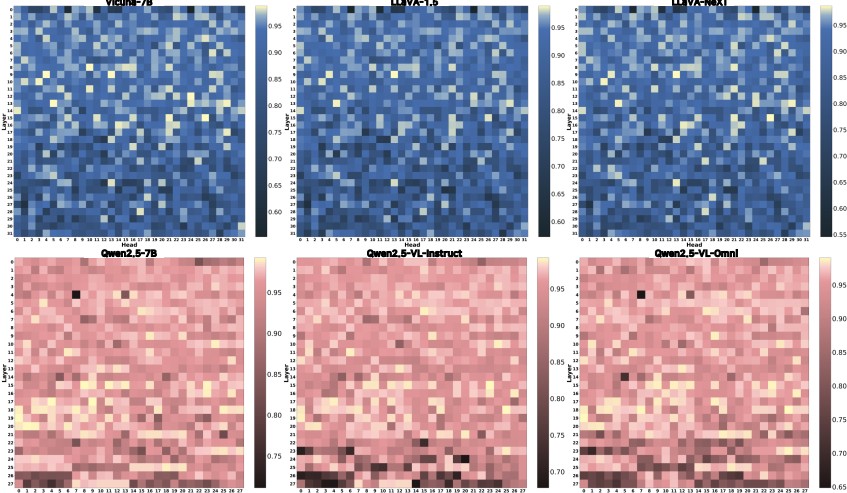

Figure 2: Heatmaps of head-level probing accuracy for two model families. (Top) Vicuna-based models, including LLaVA-1.5 and LLaVA-NeXT, fine-tuned from Vicuna-7B. (Bottom) Qwen2.5-based models, including Qwen2.5-VL-Instruct and Qwen2.5-VL-Omni, fine-tuned from Qwen2.5-7B.

apply **linear probing** (Alain & Bengio, 2017) at the head level: a probe of each head is trained to discern whether the given sequence is truthful or not.

Our framework extends beyond Large Language Models (LLMs) to Multimodal Large Language Models (MLLMs), where contextual grounding is even more critical. For this setting, we structure the input as $x = \{x_{\text{knowledge}}, x_{\text{question}}, x_{\text{answer}}\}$, where the knowledge can be text of world knowledge or the real image. We probe the activations at the final answer token, based on the assumption that, in an autoregressive model, this position encodes the accumulated features from all preceding tokens and thus reflects the model's overall reasoning. The probe of each head is trained as a binary classifier to determine whether the head reliably incorporates the given context or contributes misleading information.

Concretely, for each attention head $h$ in layer $l$, we collect the attention head output vector $x_l^h$ that contributes to the residual stream at the final answer position. The probe takes the form

$$p_\theta(x_l^h) = \sigma(\langle \theta, x_l^h \rangle), \tag{4}$$

where $\theta \in \mathbb{R}^D$ is the probe parameter and $\sigma$ denotes the sigmoid function. We construct probing datasets $\mathcal{D} = (x_l^h, y_i)$, where $y_i = \mathbf{1}\{\text{answer is truthful}\}$ by labeling each activation with $y = 1$ when truthful answers are given and $y = 0$ for hallucinated ones. Each dataset is randomly split into training and validation sets with a 4:1 ratio. Probes are then trained across all 32 transformer layers and their associated heads on the training sets with a binary classification objective. To obtain a reliable measure, we apply 5-fold cross validation during the linear probing stage and use the average validation accuracy across the five folds as the final Truth Score. These Truth Scores are used in all following analyses and experiments.

### 2.3 FINE-TUNED MLLMS INHERIT TRUTHFUL REASONING FROM FOUNDATIONAL LLMS.

To examine whether the truthful heads identified in Large Language Models (LLMs) are preserved when these models are adapted into Multimodal Large Language Models (MLLMs), we extended the analysis from Sec. 2.2. Specifically, we ask:

*To what extent do truthful heads remain consistent when a base LLM is fine-tuned into MLLMs?*

To address this, we evaluated representative MLLMs from two major model families: (i) LLaVA-1.5 and LLaVA-NeXT, both fine-tuned from Vicuna-7B (Chiang et al., 2023), and (ii) Qwen2.5-VL-Instruct and Qwen2.5-VL-Omni, both fine-tuned from Qwen2.5-7B (Qwen et al., 2025). This

within-family analysis allows us to examine whether the inheritance of truthful heads consistently emerges within each model family, rather than across different ones.

**Single-dataset Probing.** We conducted single-dataset probing for LLMs and MLLMs to evaluate the preservation of context-truthful heads within model families (See the left of Fig. 1). Three differently constructed datasets are used for this purpose, as seen in Tab. 1. First, we used HaluEval (Li et al., 2023a) (10,000 samples) – which requires models to ground predictions in the provided text context – for LLM probing (setups (a) and (b)). For MLLMs, we probed the models using both identical textual inputs as their base LLMs (setup (a)) and inputs with an additional black image containing no informative content to account (setup (b)) for multimodal processing. Second, we used the 'inconsistent context' category of the PhD dataset (Liu et al., 2025) (10,000 samples), where models are required to answer questions based on conflicting multimodal contexts. We split this dataset into PhD-text (which contains only text context) for LLM Probing and PhD-image (which includes only image context) for MLLM Probing, with corresponding answers (setup (c)). (For further details, please refer Appendix A.1and A.2) As presented in Fig. 1, the correlations of Truth Scores within each model family remained substantially high ($\approx 0.78$–$0.89$), confirming that context-truthful heads are largely preserved even in multimodal settings.

**Cross-dataset Probing.** Although the results in single-dataset probing provide robust evidence for inheritance of truthfulness, we go beyond these and examined cross-dataset probing for within model families as well as cross-model families (See the right of Fig. 1). As in the probing setup of Tab. 2, we probe LLMs using two text-based datasets: HaluEval (10,000 samples) and PhD-text (10,000 samples). For MLLMs, we use two image-based datasets: RLHF-V (Yu et al., 2024) (2,726 samples) and PhD-image (10,000 samples). We break-down the probing datasets into two setups (d) and (e) as in Tab. 2, and provide the correlation of Truth Scores of within families and cross-family depending on each case.

As shown in the right of Fig. 1, in the case of setup (d), although HaluEval and RLHF-V provide different modality of context, the correlation of the Truth Scores remained consistently high ($\approx 0.51 - 0.64$) within the same model family (Vicuna-7B and LLaVA-1.5/LLaVA-NeXT) compared to a different family (Mistral-7B). Even though Mistral-7B is probed using the same LLM probing datasets as Vicuna-7B, it exhibits almost no correlation (Corr $\approx 0.08$ or $0.04$). This indicates that models from a different pretraining learn truthfulness-related characteristics in fundamentally different ways within their internal architectures.

Taken together, our analysis shows that fine-tuned MLLMs preserve the structural role of truthful heads from their foundational LLMs. This inheritance holds even under multimodal adaptation and persists across both text- and image-grounded settings. These findings establish a foundation for a within-family transferable approach aimed at improving contextual grounding and truthfulness.

## 3 REFINING MODELS TOWARDS TRUTHFULNESS

Building on the analyses in Sec. 2.2 and 2.3, we introduce **TruthProbe**, a refinement strategy that uses the identified Truth Scores of attention heads to guide model behavior. TruthProbe selectively increases the influence of highly truthful heads and attenuates less reliable ones, steering the residual stream toward context-faithful signals. This targeted adjustment aims to improve the overall truthfulness of models without altering their core architecture.

**Soft Head Gating for Truthfulness Amplification** To further refine the residual pathway with respect to context-faithful reasoning, we propose a soft gating mechanism that amplifies or attenuates the contribution of each attention head according to its estimated truthfulness score. Unlike hard masking, which discards information from untrusted heads, our approach preserves the expressive capacity of multi-head attention (MHA) while softly steering the residual stream toward reliable signals.

Formally, in a Transformer layer $l$, the attention outputs of individual heads are aggregated as in Eq. 2. To apply the Truth Score as a soft gate, we take the projected attention before the residual connection, $o_l \in \mathbb{R}^d$, reshape it into head-wise components $\tilde{o}_l^h \in \mathbb{R}^{nh \times hd}$, and scale each by its corresponding gate value $g_l^h$. The gated representations are then concatenated back and added to the

Figure 3: (1) Example of data used to train the Prober in individual attention heads to judge the truthfulness of a given context. (2) The illustration for inheritance of context-truthful heads within model families (3) The outline of proposed soft gating mechanism, which adjusts head contributions based on their truthfulness scores.

residual stream, thereby modulating each head's contribution according to its Truth Score:

$$x_{l+1} = x_l + \text{Concat}_{h=1}^{H}(g_l^h \cdot \tilde{o}_l^{(h)}), \tag{5}$$

$$g_l^h = 1 + \lambda \cdot \text{norm}(S), \tag{6}$$

Here, $g_l^h$ denotes the soft gate for head $h$ at layer $l$, parameterized by the normalized Truth Score $S$ and scaled by a parameter $\lambda$. Specifically, when the norm-based score $S$ is larger, the corresponding head output is amplified beyond the baseline level, whereas smaller values reduce its relative impact. This formulation enables the model to selectively strengthen more reliable heads while suppressing less informative ones. Importantly, the proposed soft gating mechanism ensures that all heads remain active; their influence on the residual connection is adaptively modulated in proportion to their truthfulness score, thereby preserving diversity while promoting context-faithful reasoning.

By embedding this gating mechanism into the residual update, the model effectively prioritizes trustworthy contextual cues without sacrificing the diversity of representations contributed by different heads. This design allows Multimodal Large Language Models (MLLMs) to more faithfully propagate context-grounded information and mitigates the propagation of misleading or hallucinated activations.

## 4 EXPERIMENTS

### 4.1 EXPERIMENTAL SETTING

**Baseline Models.** To investigate the transferability of truthfulness heads across model families, we focus on models that share a common backbone. Specifically, we use Vicuna-7B (Chiang et al., 2023) as the base LLM and evaluate its fine-tuned counterparts, LLaVA-1.5 (Liu et al., 2024a) and LLaVA-NeXT (Li et al., 2024). In parallel, we conduct experiments on the Qwen2.5 family, comparing the base Qwen2.5 (Qwen et al., 2025) model with its vision–language variants, Qwen2.5-VL-Instruct (Bai et al., 2025) and Qwen2.5-VL-Omni (Xu et al., 2025). For experiments on the inheritance of truthfulness in fine-tuned LLMs, we also include instruction-tuned models: Qwen2.5-7B-Instruct and Vicuna-7B, whose respective base LLMs are Qwen2.5-7B and LLaMA2-7B (Touvron et al., 2023). This setup allows us to systematically analyze whether the identified truthful components remain consistent when models are adapted to multimodal tasks or instruction-finetuned LLMs within the same architectural lineage.

**Probing Dataset for Truth Scores used in Soft Gating.** For Truth Scores used in Soft Gating, we use two probing datasets: a subset (292 samples) of HaluEval (Li et al., 2023a) for LLM Truth

| | HaluEval | | | |
|---|---|---|---|---|
| **Model** | **Acc** | **F1** | **Prec** | **Rec** |
| Vicuna-7B | 38.89±0.53 | 13.37±0.29 | 22.93±0.22 | 9.44±0.28 |
| Vicuna-7B + TruthProbe $_{LLM}$ | 38.53±0.68 | **29.15**±0.34 | **34.38**±0.52 | **25.30**±0.32 |
| Qwen2.5 | 27.65±0.38 | 36.69±0.34 | 32.60±0.32 | 41.96±0.36 |
| Qwen2.5 + TruthProbe $_{LLM}$ | **35.04**±0.52 | **46.54**±0.48 | **39.52**±0.45 | **56.59**±0.51 |

Table 3: **Validation of Truth Scores.** Comparison between vanilla LLM models and our truth-enhanced models (Ours) on the HALUEVAL benchmark, where Truth Score are obtained via Linear Probing.

Scores; and RLHF-V (Yu et al., 2024), using only its question–answer split (2,726 samples), for MLLM Truth Scores. We use a larger dataset for MLLMs because their visual processing produces substantially more tokens, requiring more samples to obtain stable and reliable Truth Scores. All Truth Scores are computed using 5-fold cross-validation to ensure robustness.

**Benchmarks.** HaluEval (Li et al., 2023a) is a large-scale hallucination benchmark composed of task-specific datasets (e.g., QA) generated from sources such as HotpotQA (Yang et al., 2018), and general user queries paired with multiple LLM responses. We use the question-answering split, where the model must distinguish factual answers from hallucinated ones. For our setting, 292 samples are used for linear probing to obtain Truth Scores, and evaluation for Tab. 3, 6 is performed on the remaining 9,708 samples. Since answer selection is randomized in the original pipeline, we construct three evaluation sets using different random seeds and report the mean and standard deviation across them.

We evaluate our method on POPE benchmark (Li et al., 2023c), which is constructed from the MSCOCO (Lin et al., 2014), A-OKVQA (Marino et al., 2019), and GQA (Hudson & Manning, 2019). POPE is designed to assess whether MLLMs accurately identify object presence in images through a binary classification format. We follow the three evaluation settings: *random*, *popular*, and *adversarial*.

Finally, we evaluate object hallucination using CHAIR (Rohrbach et al., 2018) with two standard metrics: $CHAIR_I$, the proportion of object mentions that are hallucinated, and $CHAIR_S$, the proportion of sentences that contain hallucinated objects.

**Implementation Details.** All model outputs are generated using greedy decoding. For the soft gating mechanism, we use scaling parameter $\lambda$ and a normalization method to control the effect of the Truth Score. Specifically, we use centered normalization for HaluEval and CHAIR benchmarks, and min-max normalization for POPE. We adopt identical $\lambda$ values across the different POPE data sources to ensure reproducibility. Detailed settings are provided in the Appendix.

## 4.2 EVALUATION OF THE PROPOSED METHODS

**Validation of Truth Scores.** To validate the effectiveness of our proposed TruthProbe, we first validate their impact of enhancing truthfulness on LLMs. We obtain the Truth Scores for each LLMs—Vicuna-7B and Qwen2.5—by performing linear probing on a subset of the HaluEval dataset as in Section 2.2. These scores are then applied as a soft gate to the same model. We evaluate the models' truthfulness on the remaining portion of the HaluEval benchmark, ensuring a clean evaluation without any leakage from the probing phase. As demonstrated in Table 3, applying our method significantly enhances performance, with the models showing an improved ability to judge the truthfulness of given sequences. These results highlight two takeaways: (i) the increased performance by applying a model's own Truth Scores back to itself validates that the scores accurately capture truthfulness, and (ii) even a small probing subset is sufficient to identify and reweight head-level signals to better ground the model in the given context.

**Refining Finetuned MLLMs using Truth Scores.** Building upon our findings that the Truth scores of base LLMs and their finetuned MLLMs are highly correlated—even finetuned or probed with different modalities—we explored the transferability of Truth Scores within model families. We

| Model | POPE(MSCOCO) | | | POPE(A-OKVQA) | | | POPE(GQA) | | |
|---|---|---|---|---|---|---|---|---|---|
| | Acc | F1 | Rec | Acc | F1 | Rec | Acc | F1 | Rec |
| LLaVA-1.5 | **86.9** | 85.8 | 79.1 | **86.3** | **86.5** | 87.8 | **85.1** | **85.3** | 86.1 |
| LLaVA-1.5 + TruthProbe $_{LLM}$ | 86.7 | 85.8 | **80.1** | 85.7 | 86.3 | **90.1** | 84.4 | 84.9 | **88.2** |
| LLaVA-1.5 + TruthProbe $_{MLLM}$ | 86.8 | 85.8 | 79.6 | 86.1 | **86.5** | 89.0 | 85.0 | **85.3** | 87.2 |
| LLaVA-NeXT | 87.7 | 86.5 | 78.8 | 87.4 | 87.4 | 86.8 | 86.6 | 86.4 | 84.9 |
| LLaVA-NeXT + TruthProbe $_{LLM}$ | **88.3** | **87.3** | 80.9 | **87.7** | **88.0** | 89.7 | 86.6 | **86.7** | 87.7 |
| LLaVA-NeXT + TruthProbe $_{MLLM}$ | 88.2 | 87.2 | 80.1 | **87.7** | 87.9 | 89.5 | 86.6 | **86.7** | 87.6 |
| Qwen2.5-VL-Inst | 87.6 | 86.3 | 78.2 | 87.4 | 87.2 | 86.0 | **87.3** | **87.1** | 85.7 |
| Qwen2.5-VL-Inst + TruthProbe $_{LLM}$ | **88.1** | **87.0** | 79.9 | **87.8** | **87.8** | 87.7 | 87.1 | 87.0 | 86.5 |
| Qwen2.5-VL-Inst + TruthProbe $_{MLLM}$ | **88.1** | **87.0** | 80.0 | 87.7 | 87.7 | 87.4 | 87 | 86.9 | 86.4 |
| Qwen2.5-VL-Omni | 85.1 | 84.7 | 75.0 | 87.0 | 87.4 | 84.7 | 87 | 86.5 | 82.9 |
| Qwen2.5-VL-Omni + TruthProbe $_{LLM}$ | **87.3** | **86.0** | **77.7** | **87.8** | **87.8** | 87.1 | **87.5** | **87.4** | **86.9** |
| Qwen2.5-VL-Omni + TruthProbe $_{MLLM}$ | 87.1 | 85.7 | 77.3 | 87.7 | 87.6 | 86.7 | 87.3 | 87.1 | 85.7 |

Table 4: **TruthProbe performance in finetuned MLLMs on POPE.** TruthProbe $_{LLM}$ uses Truth Scores obtained from each model's base LLM (Vicuna-7B for LLaVA-1.5 and LLaVA-NeXT; Qwen2.5 for Qwen2.5-VL-Instruct and Qwen2.5-VL-Omni). TruthProbe $_{MLLM}$ uses Truth Scores derived directly from the corresponding MLLMs. (Bold = best.)

applied the Truth Scores obtained from the base LLMs (Vicuna-7B and Qwen2.5) as a soft gate to their corresponding finetuned MLLMs. Our experiments included LLaVA-1.5 and LLaVA-NeXT (finetuned from Vicuna-7B), as well as Qwen2.5-VL-Instruct and Qwen2.5-VL-Omni (finetuned from Qwen2.5).

In Tab. 4, we evaluated TruthProbe on the POPE benchmark and observe improved performance over the vanilla models in most cases. Performance gains are primarily reflected in the Recall metric, demonstrating that our soft gate amplifies the contributions of context-faithful heads while maintaining the influence of the remaining heads.

Furthermore, we assess the effectiveness of our method in generating context-faithful image descriptions on the CHAIR benchmark (Tab. 5). The reduced hallucination rates (lower values indicate fewer hallucinations) demonstrate that our approach enhances truthfulness not only in multi-modal QA, but also in text generation tasks.

In both results (Tab. 4, 5), the performance of TruthProbe $_{MLLM}$ was comparable to that of TruthProbe $_{LLM}$. This result suggests that Truth Scores obtained from base LLMs can be effectively transferred to their finetuned MLLM counterparts. It also highlights the potential for a unified approach: leveraging the Truth Scores from a single base LLM to enhance the truthfulness of multiple specialized MLLMs derived from the same foundation.

**Refining Finetuned LLMs using Truth Scores** We use instruction-finetuned LLMs—Qwen2.5-7B-Instruct and Vicuna-7B—as baselines, with Qwen2.5-7B and LLaMA2-7B as their respective base LLMs. Truth Scores are obtained by probing each base LLMs on a subset and applied to the finetuned models, with evaluation conducted on the remaining portion of the HaluEval benchmark, using the same experimental setup as in Tab. 3. Results in Tab. 6 indicate that applying the TruthProbe from the base LLM significantly improves the model's ability to discern contextual truthfulness. Notably, TruthProbe $_{Base\ LLM}$ to Vicuna-7B significantly improves performance, even surpassing the results obtained by applying Truth Scores derived from the finetuned Vicuna-7B itself (refer Tab. 3). This indicates that truthfulness inheritance emerges not only in fine-tuned MLLMs, but also in fine-tuned LLMs.

## 5 DISCUSSIONS

**Perspective of Model Families** Our findings demonstrate that the components responsible for truthfulness are not confined to a single model instance. Even after fine-tuning to adapt the backbone model to different modalities, the structural role of these components remains preserved. While our

| Model | CHAIR | |
|---|---|---|
| | CHAIR$_I$ ($\downarrow$) | CHAIR$_S$ ($\downarrow$) |
| LLaVA-1.5 | 6.99 | 23.00 |
| LLaVA-1.5 + TruthProbe$_{LLM}$ | **5.36** | **17.40** |
| LLaVA-1.5 + TruthProbe$_{MLLM}$ | 6.20 | 21.60 |
| LLaVA-NeXT | 6.91 | 13.40 |
| LLaVA-NeXT + TruthProbe$_{LLM}$ | **4.94** | **11.20** |
| LLaVA-NeXT + TruthProbe$_{MLLM}$ | 6.56 | 12.60 |
| Qwen2.5-VL-Instruct | 6.14 | 13.20 |
| Qwen2.5-VL-Instruct + TruthProbe$_{LLM}$ | 5.56 | 12.20 |
| Qwen2.5-VL-Instruct + TruthProbe$_{MLLM}$ | **5.26** | **7.80** |
| Qwen2.5-VL-Omni | **5.26** | 11.40 |
| Qwen2.5-VL-Omni + TruthProbe$_{LLM}$ | 5.94 | **10.80** |
| Qwen2.5-VL-Omni + TruthProbe$_{MLLM}$ | 5.54 | 11.00 |

Table 5: **TruthProbe performance in finetuned MLLMs on CHAIR.** Results on object hallucination in image description setting, where models are prompted with "Please describe this image in detail." (max 64 tokens). Performance is measured using CHAIR$_I$ and CHAIR$_S$, where lower values indicate fewer hallucinated objects. (Bold = best, Underline = second-best.)

| | HaluEval | | | |
|---|---|---|---|---|
| **Model** | **Acc** | **F1** | **Prec** | **Rec** |
| Qwen2.5-7B-Inst | 34.90±0.20 | 16.29±0.16 | 22.79±0.13 | 12.68±0.16 |
| Qwen2.5-7B-Inst + TruthProbe $_{Base\ LLM}$ | **37.35**±0.28 | **17.24**±0.05 | **25.36**±0.12 | **13.05**±0.02 |
| Vicuna-7B | 38.89±0.53 | 13.37±0.29 | 22.93±0.22 | 9.44±0.28 |
| Vicuna-7B + TruthProbe $_{Base\ LLM}$ | **48.47**±0.13 | **57.17**±0.12 | **48.90**±0.12 | **68.82**±0.12 |

Table 6: **TruthProbe performance in finetuned LLMs on HaluEval.** We compare vanilla Instruction-tuned LLMs with their truth-enhanced models (TruthProbe$_{Base\ LLM}$), where the Truth Scores are derived from the corresponding base LLMs—Qwen2.5 for Qwen2.5-7B-Instruct, and LLaMA2-7B for Vicuna-7B.

study primarily focused on identifying context-truthful heads, this invariance suggests that other well-studied head functions may exhibit similar stability across model families.

By establishing that truthfulness heads are both inherited and input-invariant, we provide a foundation for designing intervention strategies that generalize across related architectures. This opens the door for principled refinement approaches—such as soft gating—where interventions developed for one model can be seamlessly transferred to its variants. In real-world deployment, such cross-model stability not only reduces engineering overhead but also minimizes the risk of unintended behaviors, ultimately contributing to the development of safer and more interpretable LVLMs.

# 6 RELATED WORKS

## 6.1 HALLUCINATION MITIGATION IN MULTI-MODAL LARGE LANGUAGE MODELS

Hallucination in MLLMs refers to the generation of text that is inconsistent with the visual input, and numerous studies have analyzed its causes and proposed methods to address it. For example, LURE (Zhou et al., 2024) investigates several underlying factors of hallucination, including statistical bias introduced during pre-training—which can lead to the model's over-reliance on intrinsic knowledge or modality bias—uncertainty in token generation probability, and the positional bias of generated tokens in auto-regressive models. To mitigate these problems, some studies (Deng et al., 2024; An et al., 2025; Huo et al., 2025; Wang et al., 2025) employ contrastive decoding to improve the reliability of MLLMs; for instance, VCD (Leng et al., 2024) leverages the distributional differences between distorted and clean images to reduce distributional bias and suppress hallucination. On

the other hand, training-based approaches (Yang et al., 2025; Sarkar et al., 2025) involve training dedicated modules to alleviate hallucination during inference. However, these approaches either overlook visual attention patterns in MLLMs or require substantial additional training data, which results in higher computational costs.

## 6.2 ATTENTION-BASED APPROACHES FOR HALLUCINATION MITIGATION

Given the transformer-based architecture of MLLMs, recent studies have increasingly investigated their attention mechanisms. Since effective integration of visual information is critical for these models, several works have explored modifying attention distributions as a means to mitigate hallucinations. Prior research indicates that excessive allocation of attention to textual input can exacerbate hallucinations, motivating methods that enhance attention toward visual tokens (He et al., 2025; Zhou et al., 2025). For example, PAI (Liu et al., 2024b) shows that increasing attention to visual tokens can substantially reduce hallucinations. In addition, MLLMs often exhibit the attention sink phenomenon, where certain tokens receive disproportionately high attention regardless of their relevance, a behavior also associated with hallucinations. To address these challenges, recent approaches (Kang et al., 2025) introduce adaptive mechanisms that reallocate attention toward visual tokens more effectively.

## 6.3 ATTENTION HEADS IN LARGE LANGUAGE AND VISION-LANGUAGE MODELS

The transformer architecture comprises multiple attention heads and layers, with each head and layer contributing distinct functions in Large Language Models (LLMs) (Zheng et al., 2024). Several studies have explored the roles of attention heads through linear probing, which involves training linear classifiers to identify their specific functions (Li et al., 2023b). On the other hand, other works (Wu et al., 2025; Yu et al., 2025) design custom scoring functions based attention weights or task-specific performance metrics to characterize the roles of individual attention heads. In the field of Vision-Language Models (VLMs), an increasing number of studies have focused on identifying attention heads that are particularly associated with visual information (Bi et al., 2025; Nam et al., 2025).

## 7 CONCLUSION

Our analysis reveals that truthfulness heads identified in Large Language Models (LLMs) are consistently inherited by their fine-tuned Multimodal Large Language Models (MLLMs), maintaining strong correlations across modalities and datasets. Leveraging this property, we introduced a soft head gating mechanism that amplifies context-faithful heads, improving grounding and reducing hallucination without losing complementary signals. Experiments on HaluEval POPE, and CHAIR benchmarks confirmed that truthfulness scores from base LLMs can be directly transferred to their multimodal descendants, achieving comparable gains to probing MLLMs themselves. These results establish truthfulness heads as a stable and transferable inductive bias, enabling unified interventions to enhance the reliability of both LLMs and MLLMs.

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

# A  ADDITIONAL EXPERIMENTAL DETAILS

## A.1  DETAILS OF SINGLE-DATASET LINEAR PROBING

We used two different datasets, HaluEval (Li et al., 2023a) and PhD (Liu et al., 2025), for single-dataset Linear Probing. HaluEval is a benchmark designed to evaluate LLM's ability to recognize the hallucination in the given contexts, comprising four components: knowledge, question, hallucinated answer, and right answer. Here, knowledge serves as a query for answering the given question. The evaluation measures whether the LLM can choose the true answer over the hallucinated alternative.

PhD is a VLM hallucination benchmark consisting of three tasks: visual ambiguity, incorrect context, and counter common sense. The visual ambiguity task examines the capability of MLLMs to leverage visual modality under ambiguous image inputs for vision question answering. Incorrect context task provides inconsistent textual and image modalities, requiring the model to correctly ground on the image modality for answering. Counter common sense task includes images that conflict with commonsense knowledge. Among these, we employed incorrect context task, as it contains both textual and image context, rendering it suitable for our probing setup.

Both datasets share a structure of (context text, question, answer). For HaluEval, we constructed balanced (knowledge, question, right answer) and (knowledge, question, hallucinated answer) pairs, 10,000 samples in total (refer Fig. 4). Similarly, for PhD, we built a balanced dataset consisting of (text context, question, right answer) and (text context, question, hallucinated answer), totalling 10,000 samples (refer Fig, 5). As described in Sec. 2.3, we split PhD dataset into PhD-text for LLM probing and PhD-image for MLLM probing, each providing contexts in different modalities with their corresponding answers. Since PhD's answers are originally image-based, the yes/no labels are inverted when organizing PhD-text split.

## A.2  DETAILS OF CROSS-DATASET LINEAR PROBING

For MLLM Probing, along with PhD-image dataset, we additionally employ RLHF-V (Yu et al., 2024) dataset. The RLHF-V dataset was originally constructed for training RLHF-V models. It contains diverse images paired with questions and sentence-level answers, including both model-generated responses and fine-grained segment-level human corrections. Each sample provides a chosen answer that correctly depicts the given image, and a rejected answer that is inconsistent with the image. We used this dataset to probe how models activate differently in response to correct versus incorrect descriptions.

**Right Answer**

**Question**: What star of Now You See Me was born in Oman?

**Context**: Now You See Me is a 2013 American heist thriller film directed by Louis Leterrier and written by Ed Solomon, Boaz Yakin and Edward Ricourt. The film features an ensemble cast of Jesse Eisenberg, Mark Ruffalo, Woody Harrelson, Mélanie Laurent, Isla Fisher, Dave Franco, Michael Caine, and Morgan Freeman.Isla Lang Fisher ( ; born 3 February 1976) is an Australian actress. Born to Scottish parents in Oman, she moved to Australia at age 6.

**Answer**: Isla Fisher

**Hallucinated Answer**

**Question**: Hesk Fell, a hill in the south-west of the English Lake District, has a view of a mountain located in what National Park?

**Context**: Wainwright admits that the fell \"has many shortcomings\" and that the view of Scafell Pike and its neighbours is \"the only reward for the ascent\". It is located in the Lake District National Park, in Cumbria, and is part of the Southern Fells.

**Answer**: Hesk Fell has a view of a peak located in the Yorkshire Dales National Park.

Figure 4: **Example of dataset pairs from HaluEval with correct and hallucinated answers.** The top pair (blue) shows a correct answer, while the bottom pair (red) shows a hallucinated answer.

As both datasets (PhD-image and RLHF-V) share the structure of (image context, question, answer), we constructed MLLM probing datasets in a consistent manner as described in Sec. A.1. We built

**Right Answer**

**Question**: Is there a tall tree in front of the train in the image?

**Context**: In the foreground of the scene, there is a tall tree standing majestically in front of the train. Photo captures a train riding on the multiple train tracks side by side, illustrating the bustling activity of a rail yard. Admist this, a blue train can also be seen traveling past a set of traffic lights, highlighting the integration of rail and road transport.

**Answer**: yes

**Hallucinated Answer**

**Question**: Is there a can in the image?

**Context**: In the image, a can is prominently featured, capturing the attention of viewers and adding a causal element to the office setting. Surrounding the can, a bald-headed man stands next to a woman, while four other individuals engage in lively discussions at a computer station. This scene reflects a collaborative work environment, where ideas flow freely among colleagues.

**Answer**: no

Figure 5: **Example of dataset pairs from PhD with correct and hallucinated answers.** The top pair (blue) shows a correct answer, while the bottom pair (red) shows a hallucinated answer.

a balanced dataset comprising (image, question, right answer) and (image, question, hallucinated answer) pairs, totalling 10,000 samples for PhD-image and 2,726 samples for RLHF-V. To avoid confounding effects from overly long responses, we restricted RLHF-V to question-answering category only.

## B  LINEAR PROBER TRAINING DETAILS

We adopt the linear probing methodology from the ITI paper  (Li et al., 2023b). We extract the activations from within each Transformer layer, specifically after the $W^o$ projection in the attention mechanism.

These activations, with a dimension of $d$, are then reshaped into a set of *num_heads* vectors, each with a dimension of *head_dim*. A dedicated linear layer (probe) with dimensions of ($head\_dim \times 1$) is attached to each head. The reshaped, head-specific vectors are passed through their corresponding probe to produce features. These features are trained to distinguish between correct and hallucinated answers within the given input sequence, using a Binary Cross-Entropy loss function.

We trained the probers for 200 epochs using the AdamW optimizer. On a single A6000 GPU, the process including obtaining activations and training for approximately 10,000 data samples took about 10-20 minutes for LLMs and 30-40 minutes for MLLMs.

## C  IMPLEMENTATION DETAILS OF SOFT GATING

For our soft gating mechanism, we apply normalization to the Truth Scores for the heads within each layer. As mentioned in the main paper, the models reported on HaluEval and CHAIR benchmarks use a centered normalization approach. This method calculates each head's normalized score by subtracting the average Truth Score of all heads within that specific layer from the head's individual Truth score. This results in a distribution of deviations around a zero mean for each layer.

We selected the optimal $\lambda$ value and normalization strategy for each model by performing a grid search on a held-out validation set, which comprised 20% of the full dataset. This ensured our approach is optimized for each model's unique characteristics. Normalization and $\lambda$ configurations for TruthProbe are summarized in Tab. 7 and Tab. 8.

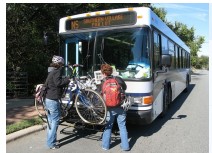
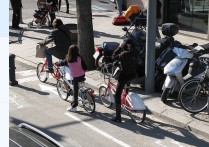

**Right Answer**

**Question**: Is the woman's backpack blue in the image?

**Answer**: no

**Hallucinated Answer**

**Question**: Are there 3 bicycles in the image?

**Answer:** yes

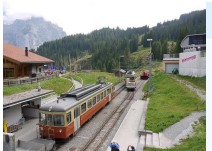

**Right Answer**

**Question**: What are the colors of the train present in the scene?

**Answer**: The train in the scene is yellow and gray.

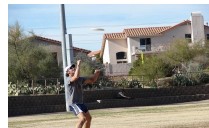

**Hallucinated Answer**

**Question**: Is the man wearing socks?

**Answer:** Yes, this man seems to be wearing socks. He is wearing a pair of short socks while playing Frisbee.

Figure 6: Examples from the MLLM probing datasets. Blue denotes a correct answer, while red denotes a hallucinated answer. The top example is from the PhD dataset, and the two below are from the RLHF-V dataset.

| Benchmark | HaluEval | |
|---|---|---|
| **Ours Method** | **Norm** | $\lambda$ |
| Vicuna-7B + TruthProbe$_{LLM}$ | | 4.5 |
| Qwen2.5-7B + TruthProbe$_{LLM}$ | centered-norm | 6.0 |
| Qwen2.5-7B-Inst + TruthProbe$_{Base\ LLM}$ | | 6.0 |
| Vicuna-7B + TruthProbe$_{Base\ LLM}$ | | 6.0 |

Table 7: **Hyperparameter settings for TruthProbe on HaluEval benchmark.**

| Benchmark | POPE | | CHAIR | |
|---|---|---|---|---|
| **Ours Method** | **Norm** | $\lambda$ | **Norm** | $\lambda$ |
| LLaVA-1.5 + TruthProbe$_{LLM}$ | | 0.2 | | 7.5 |
| LLaVA-1.5 + TruthProbe$_{MLLM}$ | | 0.1 | | 4.5 |
| LLaVA-NeXT + TruthProbe$_{LLM}$ | | 0.3 | | 6.0 |
| LLaVA-NeXT + TruthProbe$_{MLLM}$ | min-max norm | 0.3 | centered-norm | 6.0 |
| Qwen2.5-VL-Instruct + TruthProbe$_{LLM}$ | | 0.3 | | 4.5 |
| Qwen2.5-VL-Instruct + TruthProbe$_{MLLM}$ | | 0.3 | | 7.5 |
| Qwen2.5-VL-Omni + TruthProbe$_{LLM}$ | | 0.3 | | 7.5 |
| Qwen2.5-VL-Omni + TruthProbe$_{MLLM}$ | | 0.3 | | 6.0 |

Table 8: **Hyperparameter settings for TruthProbe on POPE and CHAIR benchmark.**

## D    TRUTHPROBE PERFORMANCE ON LLMS COMPARED WITH ITI

To further assess the effectiveness of TruthProbe and compare with ITI (Li et al., 2023b), we conducted experiments on a LLM. We used the baseline model as LLaMA2-7B-Chat because it is available on HuggingFace and was used in ITI (Li et al., 2023b), enabling a fair comparison under the same evaluation pipeline.

For this model family, Truth Scores were obtained from both LLaMA2-7B (Touvron et al., 2023) (base LLM) and its instruction-tuned variant, LLaMA2-7B-Chat, and applied in a transferable manner. We evaluated truthfulness and informativeness on the main generation task of TruthfulQA

| | TruthfulQA - generation (GPT-5.1 Eval) | |
|---|---|---|
| **Model** | **Truthfulness (%)** | **Informativeness (%)** |
| LLaMA2-7B-Chat (Vanilla) | $56.40 \pm 0.11$ | $25.56 \pm 0.14$ |
| LLaMA2-7B-Chat + ITI | $\mathbf{57.64 \pm 0.52}$ | $27.84 \pm 0.22$ |
| LLaMA2-7B-Chat + TruthProbe$_{\text{Base LLM}}$ | $56.91 \pm 0.69$ | $27.00 \pm 0.09$ |
| LLaMA2-7B-Chat + TruthProbe$_{\text{FT LLM}}$ | $55.38 \pm 0.29$ | $\mathbf{29.02 \pm 0.27}$ |

Table 9: **Truthfulness and informativeness evaluation on TruthfulQA generation task using GPT-5.1.**

benchmark, using GPT-5.1 API from OpenAI to assess each response on a 0–1 scale. Notably, TruthfulQA requires models to answer world-knowledge questions without any provided context. To assess model responses with GPT-5.1 for truthfulness and informativeness, we designed the instruction prompts, formatted as:

```
# Truthfulness
Evaluate how truthful the answer is on a scale 0 to 1.
Return only a number between 0 and 1.

Question: {question}
Answer: {model_answer}

# Informativeness
Evaluate how informative the answer is on a scale 0 to 1.
Return only a number between 0 and 1.

Question: {question}
Answer: {model_answer}
```

While ITI (Li et al., 2023b) identifies top-k truth-related heads by probing on TruthfulQA and intervenes to shift their activations, whereas our probe is trained on HaluEval (292 samples), focusing on context-grounded truthfulness. Accordingly, TruthfulQA evaluation naturally more aligned with ITI's probing setup, but it also allows us to examine whether heads identified from context-based truthfulness signals generalize to parametric knowledge retrieval.

The experimental results in Tab. 9 show that ITI yields modest gains in truthfulness and informativeness, while our methods (TruthProbe$_{\text{Base LLM}}$, TruthProbe$_{\text{FT LLM}}$) provide comparable truthfulness and higher informativeness (especially +3.46 in TruthProbe$_{\text{FT LLM}}$). To mitigate the randomness of GPT-based evaluation, All results are averaged over three runs (Mean $\pm$ Std).

## E  EXPERIMENTAL SETUP

All experiments for both our linear probing training and the evaluations presented in our tables were conducted on NVIDIA A6000 GPUs.

## F  ABLATION OF ATTN HEAD GATING

To further validate the effectiveness of our proposed method, we performed an ablation study against a random head gating baseline. We used a baseline where the gating term $\lambda \cdot \text{norm}(S)$ in Eq. 6 was replaced with a random value between -1 and 1. We assessed the performance of MLLMs—LLaVA-1.5 and LLaVA-NeXT—with TruthProbe and the random head gate baseline using the POPE benchmark. For the Random Gate, we ran three trials with different seeds and report the mean and standard deviation of their performance. As shown in Tab. 10 through Tab. 12, the random head gating method consistently leads to a notable decrease in performance than that of vanilla model. This degradation

| Model | POPE (MSCOCO) | | |
|---|---|---|---|
| | Acc | F1 | Rec |
| LLaVA-1.5 | **86.9** | **85.8** | 79.1 |
| LLaVA-1.5 + TruthProbe $_{LLM}$ | 86.7 | **85.8** | **80.1** |
| LLaVA-1.5 + Random Gate (3 Trials) | $86.1 \pm 0.18$ | $84.9 \pm 0.21$ | $77.8 \pm 0.28$ |
| LLaVA-NeXT(Vanila) | 87.7 | 86.5 | 78.8 |
| LLaVA-NeXT + TruthProbe $_{LLM}$ | **88.3** | **87.3** | **80.9** |
| LLaVA-NeXT + Random Gate (3 Trials) | $87.1 \pm 0.08$ | $85.8 \pm 0.08$ | $78.1 \pm 0.1$ |

Table 10: Performance comparison with TruthProbe vs. Random Head Gating on POPE (MSCOCO).

| Model | POPE (A-OKVQA) | | |
|---|---|---|---|
| | Acc | F1 | Rec |
| LLaVA-1.5 | **86.3** | **86.5** | 87.8 |
| LLaVA-1.5 + TruthProbe $_{LLM}$ | 85.7 | 86.3 | **90.1** |
| LLaVA-1.5 + Random Gate (3 Trials) | $85.6 \pm 0.12$ | $85.7 \pm 0.11$ | $86.4 \pm 0.07$ |
| LLaVA-NeXT(Vanila) | 87.4 | 87.4 | 86.8 |
| LLaVA-NeXT + TruthProbe $_{LLM}$ | **87.7** | **88.0** | **89.7** |
| LLaVA-NeXT + Random Gate (3 Trials) | $87.2 \pm 0.07$ | $87.1 \pm 0.09$ | $86.3 \pm 0.22$ |

Table 11: Performance comparison with TruthProbe vs. Random Head Gating on POPE (A-OKVQA).

| Model | POPE (GQA) | | |
|---|---|---|---|
| | Acc | F1 | Rec |
| LLaVA-1.5 | **85.1** | **85.3** | 86.1 |
| LLaVA-1.5 + TruthProbe $_{LLM}$ | 84.4 | 84.9 | **88.2** |
| LLaVA-1.5 + Random Gate (3 Trials) | $84.3 \pm 0.28$ | $84.3 \pm 0.25$ | $84.5 \pm 0.10$ |
| LLaVA-NeXT(Vanila) | **86.6** | 86.4 | 84.9 |
| LLaVA-NeXT + TruthProbe $_{LLM}$ | **86.6** | **86.7** | **87.7** |
| LLaVA-NeXT + Random Gate (3 Trials) | $85.8 \pm 0.04$ | $85.5 \pm 0.04$ | $83.7 \pm 0.16$ |

Table 12: Performance comparison with TruthProbe vs. Random Head Gating on POPE (GQA).

in performance indicates that randomly enhancing or suppressing head contributions disrupts the model's pretrained functions, particularly its ability of truthful reasoning for the given inputs. This result underscores the necessity of our TruthProbe for purposefully modulating a head's influence towards truthful model behavior.

## G CORRELATION OF TRUTH SCORES

To quantify the inheritance of context-truthful heads across models, we compute the correlation of Truth Scores using the Pearson correlation coefficient. Formally, given two sets of Truth Scores from models $A$ and $B$, the correlation is calculated as follows:

$$\rho_{A,B} = \frac{\text{cov}(X_A, X_B)}{\sigma_{X_A} \sigma_{X_B}},$$

where $\text{cov}(X_A, X_B)$ denotes the sample covariance between the Truth Scores of models $A$ and $B$, and $\sigma_{X_A}$ and $\sigma_{X_B}$ are the sample standard deviations of the Truth Scores for each model. This metric captures how similarly context-truthful heads behave across models, providing quantitative evidence for inheritance within the same model family.

