# OpenReview forum: "Identifying Truthful Inheritance in Family Models and Enhancing Truthfulness"
_ICLR.cc/2026/Conference — Submitted to ICLR 2026_

### Official Review · Reviewer_GcuH · 2025-10-27

**Soundness:** 3
**Presentation:** 2
**Contribution:** 3
**Rating:** 4
**Confidence:** 3

**Summary:**

This paper investigates (1) whether the truthfulness of LLM can be inherited to the MLLM, and (2) whether it is possible to design a structure that leverages the truthfulness mechanisms to improves the MLLM's downstream performance on POPE benchmark. The experiments provided affirmative answers to both questions.

**Strengths:**

- The finding that truthfulness inherits from the LM to VLM is insightful.
- The proposed TruthProbe improves the Acc, F1, Rec scores on the three POPE subsets (MSCOCO, A-OKVQA, GQA). It's an important advance for the interpretability field to show improvements in the model performance.

**Weaknesses:**

- The Introduction asks "Do these models inherit traits like truthfulness" which indicates other traits might also be studied, but only the truthfulness is studied in this paper. I recommend updating the questions in the abstract and the introduction correspondingly.
- The TruthProbe method would benefit from a clearer explanation of details. The number of trainable parameters in the introduced soft gate could be written clearer. From my current understanding of Section 3, $S$, $\lambda$, and $g_l^h$ are the parameters. But how many of them are trained and how many of them are hyperparameters? I'd appreciate if more details are provided. Similarly, is this method applied during inference time?
- Additionally, there are a lot of typos, and I think the paper could perhaps benefit from another round of proofreading:
  - Line 193, "1" is it Table 1 or Figure 1?
  - Line 233, "by" -> "By".
  - Line 236, the LaTeX formatting is problematic here.
  - Line 274, "equation 2" -> "Equation 2".
  - It'd be great if the notations are unified. E.g., "Equation" vs "eq", "Figure" vs "Fig.", "Sec" vs "Sec."

**Questions:**

Please refer to my reviews in the above sections. Especially, clarifications about the details would be appreciated.

---

> ### Author Response · Authors · 2025-11-22
>
> > **W1**: The Introduction asks "Do these models inherit traits like truthfulness" which indicates other traits might also be studied, but only the truthfulness is studied in this paper. I recommend updating the questions in the abstract and the introduction correspondingly.
>
> **Ans**: We thank the reviewer for the constructive feedback concerning the clarity of writing.
> We agree that the original phrasing 'traits like truthfulness' may imply a broader scope than intended. Our analysis focuses exclusively on the **truthfulness mechanism** and proposing a methodology to enhance truthfulness based on our **inheritance finding.**
> We have revised the writing in our revision to accurately reflect this focus.
>
> > **W2**: The number of trainable parameters in the introduced soft gate could be written clearer.
> From my current understanding of Section 3, S, lambda, and g_l^h are the parameters. But how many of them are trained and how many of them are hyperparameters?
> I'd appreciate if more details are provided. Similarly, is this method applied during inference time?
>
> **Ans**:
> The training of new parameters occurs **only during the Linear Probing stage**.
>
> **Trainable Parameters**
>
> A linear layer probe ($\theta_l^h$) is trained for each attention head. The total number of trainable parameters in this stage is calculated as: $h_d \times n_l \times n_h$, where $h_d$ is the head dimension (e.g., 128 for Vicuna-7B), $n_l$ is the number of layers, and $n_h$ is the number of heads.
>
> **Refining during Inference**
>
> The Truth Score ($S_l^h$) for each head is obtained by performing Linear Probing. During model inference, these scores are kept fixed and applied with hyperparameters to adjust the attention output via the Soft Gate mechanism. The Soft Gate itself contains no trainable parameters. The method has two hyperparameters: the scaling factor $\lambda$ and the specific normalization method applied to the $S_l^h$ scores. (For further details, please refer to Sec. 4.1 and Appendix C.)
>
> > **W3**: Additionally, there are a lot of typos, and I think the paper could perhaps benefit from another round of proofreading
>
> **Ans**: We sincerely thank the reviewer for their meticulous proofreading. We have conducted a thorough review of the manuscript, and we standardized the notations and reflected the detailed clarification for method and experiments.
>
> > **Q**: Please refer to my reviews in the above sections. Especially, clarifications about the details would be appreciated.
>
> **Ans**: We have incorporated all the feedback mentioned into the revised manuscript. For further clarifications regarding our analyses and experiments, please refer to our responses to Reviewer xfE4.

---

### Official Review · Reviewer_QyjF · 2025-10-29

**Soundness:** 3
**Presentation:** 2
**Contribution:** 1
**Rating:** 2
**Confidence:** 4

**Summary:**

Previous work ITI shows that some attention heads in LLMs are highly associated with truthfulness, as they can distinguish the truthfulness of an input with high accuracy in linear probing. Following ITI, this paper finds that these truthfulness-associated heads are inherited in the MLLMs fine-tuned from a base LLM. They then propose a representation steering method to enhance the truthfulness of the MLLMs.

**Strengths:**

1. The authors pose the research questions clearly in the paper.

**Weaknesses:**

1. It seems that the first clarified contribution (Line 092) is not a contribution of this paper, as this is achieved in ITI. The paper simply applies ITI on Halu dataset and clarifies that the identified heads are context-truthful heads, not truthful heads.
2. One contribution of this paper is the finding on inheritance of truthfulness. However, how to derive the correlation scores that validate this finding is not explicitly clarified in Section 2.3. That is, only scores are given in Lines 236-240, while the definition or derivation of these scores are not clarified.
3. Overall, I feel this work is an application of ITI on the Halu dataset and MLLMs.

**Questions:**

1. In the experiment section, the authors show the effectiveness of the proposed TruthProbe on the fine-tuned MLLMs. Since the proposed method is designed to improve truthfulness for general LLMs/MLLMs, I wonder if there is any result showing the effectiveness on the base LLMs, especially compared with ITI?
2. This paper shows that the truthfulness inheritance happens on fine-tuned MLLMs. I am curious if this also happens on the fine-tuned LLMs.

---

> ### Author Response · Authors · 2025-11-22
>
> > **W1**: W1: It seems that the first clarified contribution (Line 092) is not a contribution of this paper, as this is achieved in ITI. The paper simply applies ITI on Halu dataset and clarifies that the identified heads are context-truthful heads, not truthful heads.
> > **W3**: Overall, I feel this work is an application of ITI on the Halu dataset and MLLMs.
>
> **Ans**: We appreciate the reviewer’s observation and acknowledge that our probing procedure builds on the methodology introduced by ITI. However, our contributions go beyond a direct application: we (i) **identify the inheritance of truthful heads within model families**, and (ii) demonstrate that applying *Soft Gating* on these heads yields a **transferable mitigation approach within the same family**. We have revised the paper to make sure the probing method is not interpreted as our own contribution, and we have clarified our actual contributions more carefully.
>
> > **W2**: One contribution of this paper is the finding on inheritance of truthfulness. However, how to derive the correlation scores that validate this finding is not explicitly clarified in Section 2.3. That is, only scores are given in Lines 236-240, while the definition or derivation of these scores are not clarified.
>
> **Ans**: We thank the reviewer for pointing this out. We compute the correlation using the Pearson correlation coefficient.
> Formally, the correlation is calculated as:
> $$
> r_{xy} = \frac{\mathrm{Cov}(x, y)}{s_x s_y},
> $$
> where $\mathrm{Cov(x,y)}$ denotes the sample covariance between the two sets of Truth Score for Model $x$ and Model $y$, and $s_x$ and $s_y$ represent their sample standard deviations of the Truth Score for each model, respectively.
>
> We have included the derivation for correlation of Truth Scores in our revision (Appendix G).

---

> ### Author Response · Authors · 2025-11-22
>
> > **Q1**: In the experiment section, the authors show the effectiveness of the proposed TruthProbe on the fine-tuned MLLMs. Since the proposed method is designed to improve truthfulness for general LLMs/MLLMs, I wonder if there is any result showing the effectiveness on the base LLMs, especially compared with ITI?
>
> **Ans**: We also evaluated the proposed method on a base LLM, specifically LLaMA2-7B-Chat, to examine whether the **model-family transferability of our Soft Gate holds for LLMs** as well. We selected LLaMA2-7B-Chat (which is one of the HF-available models provided by ITI) to enable a fair comparison using the same evaluation pipeline.
>
> For this model family, we obtained Truth Scores from both LLaMA2-7B (base) and LLaMA2-7B-Chat (its instruction-tuned variant), and applied our TruthProbe in a transferable manner. We evaluated *truthfulness* and *informativeness* on the main generation task of TruthfulQA benchmark, using GPT-5.1 to assess each response on a 0–1 scale. (TruthfulQA requires models to answer world-knowledge questions without any provided context.)
>
> ITI identifies top-k truth-related heads by probing on TruthfulQA and intervenes to shift activations on those heads, whereas our probe is trained on HaluEval (292 samples), focusing on context-grounded truthfulness. Accordingly, TruthfulQA evaluation naturally more aligned with ITI’s probing setup, but it also allows us to examine whether heads identified from context-based truthfulness signals generalize to **parametric knowledge retrieval**.
>
> **| TruthfulQA (generation) |**
> |  | GPT Eval - Truthfulness | GPT Eval - Informativeness |
> |---|---|---|
> | **LLaMA2-7B-Chat (Vanilla)** | 56.40±0.11 | 25.56±0.14 |
> | **LLaMA2-7B-Chat + ITI** | **57.64±0.52** | 27.84±0.22 |
> | **LLaMA2-7B-Chat + TruthProbe_Base LLM** | 56.91±0.69 | 27.00±0.09 |
> | **LLaMA2-7B-Chat + TruthProbe_FT LLM** | 55.38±0.29 | **29.02±0.27** |
>
> Experimental results show that ITI yields modest gains in truthfulness and informativeness, while our methods (TruthProbe_Base LLM, TruthProbe_FT LLM) provide comparable truthfulness and higher informativeness (especially +3.46 in TruthProbe_FT LLM). To mitigate the randomness of GPT-based evaluation, All results are averaged over three runs (Mean ± Std).

---

> ### Author Response · Authors · 2025-11-22
>
> > **Q2**: This paper shows that the truthfulness inheritance happens on fine-tuned MLLMs. I am curious if this also happens on the fine-tuned LLMs.
>
> **Ans**: Thank you for the insightful suggestion. We conducted an additional experiment to examine **whether truthfulness inheritance also appears in fine-tuned LLMs**. Specifically, we used Qwen2.5-7B and LLaMA2-7B as base models, and their instruction-tuned counterparts Qwen2.5-7B-Instruct and Vicuna-7B.
>
> For each base LLM (Qwen2.5-7B, LLaMA2-7B), we performed probing to obtain the Truth Scores. We then applied the resulting Soft Gate to the corresponding fine-tuned LLM (Qwen2.5-7B-Instruct, Vicuna-7B) and evaluated the performance changes on HaluEval benchmark.
>
> **| HaluEval |**
>
> HaluEval results are averaged over three random seeds for answer selection, as detailed in our response to W4 by Reviewer xfE4.
>
> |  | Acc | F1 | Prec | Rec |
> |---|---|---|---|---|
> | **Qwen2.5-7B-Instruct** | 34.90±0.20 | 16.29±0.16 | 22.79±0.13 | 12.68±0.16 |
> | **Qwen2.5-7B-Instruct + TruthProbe_Base LLM (Qwen2.5-7B)** | **37.35±0.28** | **17.24±0.05** | **25.36±0.12** | **13.05±0.02** |
> | **Vicuna-7B** | 38.89±0.53 | 13.37±0.29 | 22.93±0.22 | 9.44±0.28 |
> | **Vicuna-7B + TruthProbe_Base LLM (LLaMA2-7B)** | **48.47±0.13** | **57.17±0.12** | **48.90±0.12** | **68.82±0.12** |
>
> As shown in the table, applying the TruthProbe from the base LLM significantly improves the model’s ability to discern contextual truthfulness. This indicates that truthfulness inheritance emerges not only in fine-tuned MLLMs, but also in fine-tuned LLMs, further strengthening the claim of the paper.

---

### Official Review · Reviewer_xfE4 · 2025-11-01

**Soundness:** 3
**Presentation:** 2
**Contribution:** 2
**Rating:** 4
**Confidence:** 4

**Summary:**

In this paper, the authors investigate whether truthfulness mechanisms in large language models (LLMs) are inherited by their multimodal counterparts (MLLMs) and propose an intervention technique to enhance truthfulness performance. The authors employ linear probing at the attention head level to obtain truthfulness scores that indicate each head's responsiveness to truthful outputs. The authors then compute correlation coefficients between the truthfulness scores (probe accuracies) of base LLMs and their fine-tuned MLLMs to demonstrate inheritance. Building on these findings, they propose TruthProbe, a soft gating mechanism that amplifies contributions from high-truthfulness heads during inference by scaling their outputs in the residual stream according to normalized truthfulness scores. In the experiments, the authors evaluate the intervention method on the HaluEval benchmark for LLMs and the POPE benchmark for MLLMs.

**Strengths:**

- The paper presents an innovative perspective on truthfulness mechanisms and intervention techniques. The inheritance framework provides a novel angle to understand truthful behavior, and the findings from interpretability results inform the design of the intervention mechanism.

- The method is evaluated across multiple benchmarks and experimental settings, providing a comprehensive view of the proposed approach.

**Weaknesses:**

- The presentation of the paper could be improved to enhance clarity and include necessary methodological details.
  - The paper lacks clarity on the dataset splitting strategy for Table 1 evaluation. While the authors mention a 4:1 train/validation split for probe training and use a held-out validation set (20%) for hyperparameter tuning, it remains unclear whether Table 1 evaluates on truly independent test data or data potentially overlapping with the probe training/validation sets.
  - In Figure 3, the evaluation results are obtained by aggregating across three benchmarks, but the paper does not specify how the aggregation is performed or provide breakdown scores for each individual benchmark.
  - Figure 2 presents truthfulness score breakdowns for each attention head across all layers, but the heatmap format offers a low information-to-noise ratio.

- The rationale and evidence should be provided to support the probing applied at the final answer token position, as it may not be the place where untruthfulness manifests.

- The methodology for evaluating truthfulness inheritance in MLLMs has limitations that weaken the paper's core claims.
  - Evaluating MLLMs' truthfulness mechanisms primarily through text-based hallucination benchmarks (HaluEval) with blank images does not provide meaningful insights into multimodal truthfulness. This evaluation setting does not reflect typical MLLM use cases, where models process informative visual content alongside text. The approach conflates text-only reasoning capabilities inherited from the base LLM with genuine multimodal grounding abilities.
  - The observed decrease in similarity when transitioning from text-only to multimodal settings (Figure 1(b) vs. Figure 3) may actually suggest that MLLMs develop new or altered truthfulness mechanisms distinct from their LLM counterparts, contradicting the inheritance hypothesis rather than supporting it.
  - The cross-modal similarity comparison does not constitute strong evidence for inheritance. The authors compare models trained from scratch (Mistral) with fine-tuned derivatives to demonstrate family-specific patterns, but this contrast is insufficient. Different base models trained on different datasets with different procedures naturally exhibit different internal representations. The low correlation with Mistral (≈0.02) could simply reflect differences in pre-training rather than validating that fine-tuning preserves specific functional properties.

- The experimental results show marginal performance differences between intervened and non-intervened models (Table 2 and Figure 5), raising questions about the method's practical effectiveness. To establish that the intervention produces statistically significant improvements rather than noise-level fluctuations, the authors should report results from multiple runs with different random seeds and conduct appropriate statistical significance tests.

**Questions:**

Please see the above section.

---

> ### Author Response · Authors · 2025-11-22
>
> > **W4**: The experimental results show marginal performance differences between intervened and non-intervened models (Table 2 and Figure 5), raising questions about the method's practical effectiveness. To establish that the intervention produces statistically significant improvements rather than noise-level fluctuations, the authors should report results from multiple runs with different random seeds and conduct appropriate statistical significance tests.
>
> **Ans**: Thanks for raising the important point. We first clarify that both our POPE and HaluEval evaluation utilizes greedy decoding, which is a deterministic process; as a result, the output is independent of the model’s random seed.
>
> To address the concern regarding ‘noise-level fluctuations’ and ensure statistical robustness and reliability, we implemented 5-fold Cross-Validation (5-fold CV) during Linear Probing stage to obtain Truth Score for POPE and HaluEval evaluation.
> Specifically, all presented results, including the correlation between Truth Scores (from Linear Probing) across different models and the final benchmark evaluation outcomes, have been updated using the results averaged over the 5-Fold CV during the Linear Probing.
>
> Additionally, we report the performance metrics in Tab.1 (HaluEval) as the (Mean ± Std). This reflects our approach of constructing the HaluEval benchmark (which measures the model’s ability to discern the answer’s truthfulness given ‘knowledge + question + answer’) using three different random seeds for selecting between the right and hallucinated answers. This process demonstrates the statistical stability of our evaluation.
>
> Furthermore, to demonstrate the practical effectiveness of our method, we additionally show its impact on benchmarks that evaluate the mitigation of object hallucination during image-based text generation (CHAIR [1]). Please refer to the Global Response for our results on CHAIR.
>
>
> > **W3-1**: Evaluating MLLMs' truthfulness mechanisms primarily through text-based hallucination benchmarks (HaluEval) with blank images does not provide meaningful insights into multimodal truthfulness. This evaluation setting does not reflect typical MLLM use cases, where models process informative visual content alongside text. The approach conflates text-only reasoning capabilities inherited from the base LLM with genuine multimodal grounding abilities.
>
> **Ans**: We thank the reviewers for raising the necessity of evaluating genuine multimodal grounding abilities in MLLMs.
>
> To address this and **better reflect MLLM use cases in our single-dataset probing** (Fig.1), we introduce new results using the ‘inconsistent context’ category of PhD dataset [2] . This category requires the models to predict the answer for the question given the conflicting multimodal (image and text) context. We further split this dataset into PhD-text (which contains only text context) for LLM Probing dataset and PhD-image (which includes only image context) for MLLM Probing, with corresponding answers. (Further details are provided in the updated Section 2.3, Appendix A.1 and A.2.)
>
> **| Correlation of Truth Scores for Vicuna-7B model families |**
> |                 | LLaVA-1.5 | LLaVA-NeXT |
> |-----------------|-------------------|--------------------|
> | **Vicuna-7B** | 0.89              | 0.88               |
>
>
> **| Correlation of Truth Scores for Qwen2.5-7B model families |**
> |                  | Qwen2.5-VL-Instruct | Qwen2.5-VL-Omni |
> |------------------|-----------------------------|-------------------------|
> | **Qwen2.5-7B** | 0.84                        | 0.78                    |
>
>
> The correlation within model families using the multimodal single-dataset (PhD) remained substantially high (≈ 0.78–0.89), providing more meaningful insights into multimodal truthfulness.
>
> [1] Rohrbach, Anna, et al. "Object hallucination in image captioning." (EMNLP, 2018)
>
> [2] Liu, Jiazhen, et al. "PhD: A ChatGPT-Prompted Visual Hallucination Evaluation Dataset." (CVPR, 2025)

---

> ### Author Response · Authors · 2025-11-22
>
> > **W1-2**: In Figure 3, the evaluation results are obtained by aggregating across three benchmarks, but the paper does not specify how the aggregation is performed or provide breakdown scores for each individual benchmark.
>
> **Ans**:
> In the right of Fig.1 (Fig.3 before revision), we probe LLMs (Vicuna-7B and Mistral-7B) using two text-based datasets: HaluEval (10,000 samples) and PhD-text (10,000 samples). For MLLMs (LLaVA-1.5 and LLaVA-NeXT), we use two image-based datasets: RLHF-V (2,726 samples) and PhD-image (10,000 samples). We have incorporated this details in our paper revision (Sec. 2.3).
>
> We break down the probing datasets into two groups (a, b), and provide the correlation of Truth Score of within families and cross-family depending on each case.
>
> | group | LLMs Probing Data | MLLMs Probing Data |
> |---|---|---|
> | a | HaluEval (10,000) | RLHF-V (2,726) |
> | b | PhD-text + HaluEval (20,000) | PhD-img + RLHF-V (12,726) |
>
> | [ group a ] | LLaVA-1.5 | LLaVA-NeXT | Mistral |
> |---|---|---|---|
> | **Corr w/ Vicuna-7B** | 0.62 | 0.64 | 0.08 |
>
> | [ group b ] | LLaVA-1.5 | LLaVA-NeXT | Mistral |
> |---|---|---|---|
> | **Corr w/ Vicuna-7B** | 0.57 | 0.51 | 0.04 |
>
> In the case of group a, although HaluEval and RLHF-V provide different modality of context (text vs. image), the correlation of the Truth Score—which measures how well each head grounds the given context—appears substantially higher within the same model family (Vicuna-7B and LLaVA-1.5/LLaVA-NeXT) compared to a different family (Mistral-7B).
>
> As shown in the table, even though Mistral-7B is probed using the same LLM probing datasets as Vicuna-7B, it exhibits almost no correlation (Corr ≈ 0.08 or 0.04). This indicates that models from a different pretraining learn truthfulness-related characteristics in fundamentally different ways within their internal architectures.
>
> > **W3-2**: The observed decrease in similarity when transitioning from text-only to multimodal settings (Fig. 1(b) vs. Fig. 3) may actually suggest that MLLMs develop new or altered truthfulness mechanisms distinct from their LLM counterparts, contradicting the inheritance hypothesis rather than supporting it.
>
> **Ans**: We acknowledge the reviewer's point regarding the decrease in correlation. However, we emphasize that the relative correlation is the key factor.  We compare this against the correlation between Vicuna and Mistral, which remains extremely low (≈ 0.08 or 0.04), despite them being probed using the same LLM data and sharing similar underlying architectures.
>
> Crucially, the LLaVA-1.5/LLaVA-NeXT pair maintains a relatively high correlation (≈ 0.57 and 0.51) in the complex multimodal setting (group b). This meaningful level of correlation, which is significantly higher than the ≈ 0.04 baseline, suggests that a substantial portion of the original LLM's truthfulness mechanism is inherited and preserved in the MLLM, even after multimodal training induces necessary changes to the final mechanism.
>
> > **W3-3**: The cross-modal similarity comparison does not constitute strong evidence for inheritance. The authors compare models trained from scratch (Mistral) with fine-tuned derivatives to demonstrate family-specific patterns, but this contrast is insufficient.
> Different base models trained on different datasets with different procedures naturally exhibit different internal representations. The low correlation with Mistral (≈0.02) could simply reflect differences in pre-training rather than validating that fine-tuning preserves specific functional properties.
>
> **Ans**: Our central contention is that a specific property (e.g., truthfulness) is established during pre-training, and the characteristic structure of this internal mechanism is preserved (inherited) throughout subsequent fine-tuning.
>
> > **W1-1**: The paper lacks clarity on the dataset splitting strategy for Table 1 evaluation. While the authors mention a 4:1 train/validation split for probe training and use a held-out validation set (20%) for hyperparameter tuning, it remains unclear whether Table 1 evaluates on truly independent test data or data potentially overlapping with the probe training/validation sets.
>
> **Ans**: For obtaining the Truth Score (which is later applied as a soft gate), we used 292 HaluEval samples for probing and the remaining 9,708 samples exclusively for LLM evaluation. We have included these details in our revision to ensure the data splitting strategy is fully transparent.
>
> > **W2**: The rationale and evidence should be provided to support the probing applied at the final answer token position, as it may not be the place where untruthfulness manifests.
>
> **Ans**: Thank you for raising this point. Our choice to probe the final answer token is based on the assumption that, in an autoregressive model, the activation at the last token encodes the accumulated features of all preceding tokens. Thus, it inherently reflects the model’s reasoning process and the contextual signals that influence truthfulness.

---

### Author Response · Authors · 2025-11-22

We thank all reviewers for thoughtful reviews and constructive feedback.

- **Reviewer xfE4** provided insightful comments on **methodology of validating truthfulness inheritance**, which helped us to strengthen our core claim. We also appreciate their suggestion to ensure **statistical robustness**, which guided us in presenting more reliable and rigorous evaluation.
- **Reviewer QyjF** provided valuable feedback on **how to clearly differentiate ITI’s probing method from our contributions**, and suggested additional experiments to demonstrate the method’s **effectiveness on LLMs**, including whether truthfulness inheritance occurs in fine-tuned LLMs.
- **Reviewer GcuH** provided detailed suggestions aimed at improving the **clarity of our methodology** and the **overall writing**. We are grateful for these suggestions and have carefully addressed them in the revised version of the manuscript.


In addition, we include new results showing that **our TruthProbe also effectively mitigates object hallucination in MLLMs** on the CHAIR [1] benchmark, demonstrating that our refinement approach for enhancing truthfulness also benefits image-grounded generation.

**| Performance on CHAIR benchmark |**

For CHAIR evaluation, we prompted the model with "Please describe this image in detail" and set  the maximum output tokens to 64. Object hallucination was measured at both the instance and sentence levels.

|  | CHAIR_i (↓) | CHAIR_s (↓) |
|---|---|---|
| **LLaVA-1.5** | 6.99 | 23.00 |
| **LLaVA-1.5 + TruthProbe_LLM** | **5.36** | **17.40** |
| **LLaVA-1.5 + TruthProbe_MLLM** | 6.20 | 21.60 |

|  | CHAIR_i (↓) | CHAIR_s (↓) |
|---|---|---|
| **LLaVA-NeXT** | 6.91 | 13.40 |
| **LLaVA-NeXT + TruthProbe_LLM** | **4.94** | **11.20** |
| **LLaVA-NeXT + TruthProbe_MLLM** | 6.56 | 12.60 |

|  | CHAIR_i (↓) | CHAIR_s (↓) |
|---|---|---|
| **Qwen2.5-VL-Instruct** | 6.14 | 13.20 |
| **Qwen2.5-VL-Instruct + TruthProbe_LLM** | 5.56 | 12.20 |
| **Qwen2.5-VL-Instruct + TruthProbe_MLLM** | **5.26** | **7.80** |

|  | CHAIR_i (↓) | CHAIR_s (↓) |
|---|---|---|
| **Qwen2.5-VL-Omni** | **5.26** | 11.40 |
| **Qwen2.5-VL-Omni + TruthProbe_LLM** | 5.94 | **10.80** |
| **Qwen2.5-VL-Omni + TruthProbe_MLLM** | 5.54 | 11.00 |

[1] Rohrbach, Anna, et al. "Object hallucination in image captioning." (EMNLP, 2018)

---

### Author Response · Authors · 2025-12-01
**Revisions based on Reviewers' Feedback**

We sincerely appreciate the constructive feedback provided by the reviewers. To thoroughly address all concerns, we conducted extensive additional analyses and experiments, and updated the manuscript accordingly.

Below is a concise summary of our revisions.

**Revisions Based on Reviewer xfE4’s Feedback**

- Strengthened the analysis for the Truthfulness Inheritance (**Sec. 2.3**) by:
    - introducing new results using the PhD dataset to better reflect real MLLM use cases in single-dataset probing (**Tab.1, Fig. 1**),
    - further breaking down the cross-dataset probing setups, and reporting an additional analysis of correlations of Truth Scores within and across model families (**Tab.2, Fig. 1**), providing stronger evidence for *inheritance*.
- Performed statistical significance tests (**5-fold CV** for Truth Scores; HaluEval and TruthfulQA evaluation **averaged over 3 runs**).
- Added clarification on the rational for probing setup (activation at final answer token) (**Sec. 2.2**) and on the datasets for both probing and evaluation (**Sec. 2.3., Sec. 4.1, Appendix A**).
- Added a more informative figure to more clearly present the inheritance phenomenon (**Fig.1**).

**Revisions Based on Reviewer QyjF’s Feedback**

- Clearly differentiated ITI’s probing scheme from our *contributions*, which identify Context-Truthful heads, uncover inheritance, and introduce a transferable approach within model families.
- Reported the effectiveness of TruthProbe on LLMs compared with ITI (**Appendix D**).
- Added new results demonstrating truthfulness inheritance in finetuned LLMs (**Tab. 6**), further extending the scope of our inheritance findings.

**Revisions Based on Reviewer GcuH’s Feedback**

- Improved writing clarity, including revisions to the research question in the *introduction* and proofreading.
- Strengthened the logic, analyses, and experiments supporting the paper’s core claims.

**Global Update.**

- Added new experiments for the practical effectiveness of TruthProbe on CHAIR (**Tab. 5**), demonstrating TruthProbe’s ability of mitigating object hallucination in image-based text generation.


We hope these revisions clearly demonstrate our efforts to address all concerns and further strengthen the manuscript.

---

### Meta-Review · Area_Chair_2uQb · 2025-12-29

**Summary:**

Paper Summary. This paper investigates if the truthfulness mechanism is inherited from a base LLM to its finetuned MLLMs. The truthfulness mechanism is based on linear probing on attention heads to obtain truthfulness scores, and the Inheritance is measured by correlation coefficients of the truthfulness scores between the base LLM and its finetuned MLLMs. The experiments demonstrate a strong correlation. Based on this, they further propose a soft gating method using the truthfulness score to enhance contextual truthfulness. Due to the inheritance of the truthfulness mechanism, the Truthfulness scores from base LLMs can be transferred and applied for gating to its finetuned MLLMs.

Paper Strengths. The paper offers an insightful perspective of inheritance of truthfulness mechanisms from LLMs to finetuned MLLMs. The paper leverages probing of truthfulness to design a transferable truthfulness enhancement method.

Reviewer Concerns. (1) The scope and contribution of this paper is not clear or significant (Reviewer GcuH, Reviewer QyjF). (2) The probing methodology is not sound. Evaluating MLLMs' truthfulness mechanisms primarily through text-based hallucination benchmarks. (Reviewer xfE4). (3) The data splitting and evaluation setup should be clarified (Reviewer xfE4). (4) The truthfulness improvement is marginal and it should report significance test (Reviewer xfE4). (5) The implementation and working principles of probing at the final answer token position and soft gating mechanisms can be further clarified (Reviewer GcuH, Reviewer xfE4). (6) Writing Typos (Reviewer GcuH).

**Reviewer Concerns:**

Addressed Concerns.
The author provided solid answers to concerns including (2) (3) (4) (5) (6)

Outstanding Concerns.
There two outstanding concerns

The first is the scope and significance of the contribution of this paper. Given that the previous work ITI already proposed linear probing to identify the truthfulness of LLMs based on attention heads, the technical novelty of this work is limited as it is a straightforward application of existing approaches. Furthermore, one contribution is to find the inheritance of truthfulness and the truthfulness enhancement gating, but the scope of this is restricted to only the finetuned MLLMs. The author provides further experiments on base LLMs and finetuned LLMs, but the results on base LLMs do not show significant improvement over ITI, and the results on finetuned LLMs lacks more baseline intervention methods.

Second, I also want to make one point that is not mentioned by reviewers. I believe the models used for experiments are largely outdated, so it is unclear whether the proposed finding and technique is still relevant to most recent models.

**Reviewer Scores:**

Reviewer xfE4: 4 -> 6

Reviewer QyjF: 2 -> 2 due to the outstanding concern (1)

Reviewer GcuH: 4 -> 6

---

### Decision · Program_Chairs · 2026-01-26

Reject